# "The only friend I had was my gun": A mixed-methods study of gun culture in school shootings

Anne Nassauer●*

Faculty of Social Sciences, Law, and Economics, University of Erfurt, Erfurt, Germany

* anne.nassauer@uni-erfurt.de

## Abstract

Firearms are the leading cause of death for minors in the United States and US gun culture is often discussed as a reason behind the prevalence of school shootings. Yet, few studies systematically analyze if there is a connection between the two: Do school shooters show a distinct gun culture? This article studies gun culture in action in school shootings. It studies if school shooters show distinct meanings and practices around firearms prior to the shooting, as well as patterns in access to firearms. To do so, I analyze a full sample of US school shootings. Relying on publicly available court, police, and media data, I combine qualitative in-depth analyses with cross-case comparisons and descriptive statistics. Findings suggest most school shooters come from a social setting in which firearms are a crucial leisure activity and hold meanings of affection, friendship, and bonding. These meanings translate into practices: all school shooters had easy access to the firearms they used for the shooting. Findings contribute to research on firearms and youth violence, public health, as well as the sociology of culture.

## Introduction

Since 2020 guns are the leading cause of death for American children and teens [1]. Each hour of each day a child is shot in the United States: in the 2010s more than 30,000 young people were killed by firearms [2]. Both the World Health Organization and the US Centers for Disease Control and Prevention [3] see gun violence as a vital public health issue.

US gun culture is described as unique, from its historical context and legal framework to the high levels of gun ownership. While research highlights that Americans live in a distinct gun culture, the term "culture" leaves much to be desired [4]. Swidler [5, p.273] describes culture as "symbolic vehicles of meaning, including beliefs, ritual practices, art forms, and ceremonies, as well as informal cultural practices such as language, gossip, stories, and rituals of daily life." Her concept of "culture in action" understands culture as more than merely a set of static values. It emphasizes that culture functions as a flexible toolkit of symbols, practices, and strategies that individuals draw on to navigate and shape their actions in specific contexts. Importantly, this understanding sees culture as dynamic and as constantly shaped through social interactions with others. But what constitutes gun culture in action? What do practices and meanings look like and what are the results of this culture? Yamane [6] suggests that an

**Data availability statement:** All relevant data are within the manuscript and its Supporting Information files.

**Funding:** The author(s) received no specific funding for this work.

**Competing interests:** The authors have declared that no competing interests exist.

ideal way to explore broader questions of culture in action may be to explore the systematic meaning of guns in the US: Analyzing them as specific objects of culture offers the opportunity to study how symbolic elements of cultural meaning are socially constructed, thus influencing practices and strategies of action. Scholars highlight that we need more research exploring gun culture in its various dimensions, including extreme gun-related incidents, like shooting sprees [4,6,7]. Further, we must discern the local symbolic meanings of guns and the sources of their semiotic power [4].

Discursively, gun culture in the US is often tied to school shootings – incidents where students randomly fire at others at their current or former school [8,9]. Yet, school shootings present a puzzle: Many people live in the US gun culture, but school shootings are an extreme and rare type of human behavior. Why would the omnipresent and often fairly abstract concept of gun culture translate into practices for the shooters, but not for most other people? We need more insights into whether gun culture actually matters for school shooters, and whether school shooters show indicators of culture in action that lead to distinct meanings or practices around guns.

This article studies this topic by discussing findings from an analysis of gun culture and firearm access in US rampage school shootings. I analyze a full sample of school shootings; incidents that result in mass shootings (four or more killed) as well as attacks where fewer people or no one was killed. I triangulate analytic steps: I carve out the role of meaning making and practices in school shooters lives through in-depth case studies, identify patterns through cross case comparisons, and study whether they translate into practices using descriptive statistics. The goal is to identify possible patterns of gun culture in action in school shootings, as well as to explore culture not merely as a broader set of static values or beliefs in the US, but as something that is actively enacted and negotiated by actors and their immediate social surroundings in everyday interactions.

Throughout, I provide specific case examples to illustrate findings. To give as much information as possible on the analyzed cases and data, I mention 24 of the 83 analyzed cases as examples. Exemplary cases were chosen across decades, US states, and perpetrator age, to highlight that identified patterns exist across the sample. All data are publicly available and numerous sources are cited in the article (see S1 Data). In S2 Data, I provide a full list of cases with information on gun culture specifics for each case. So as to not give shooters their sometimes-desired fame, I do not name any of the shooters, instead using pseudonyms throughout. However, findings can also be verified using the case information (date, location) provided in S2 Data. Ultimately, this study aims to contribute to our understanding of culture in action and the role of gun culture in US school shootings.

## Research on guns and rampages

Gun researchers note that while a range of vital dynamics have already been identified, the field still deserves more sociological attention [4,6,10]. Among existing studies on firearms, studies examining gun legislation highlight that although public support for stricter gun legislation is rising, gun laws in the US remain less restrictive than in most other countries [4]. Federal law sets minimum standards for firearm regulation, but individual states have their own laws, some stricter, others more lenient.

Research on gun ownership underlines that the US has the highest number of firearms per capita in the world and has a uniquely strong cultural association of guns with personal and national identity [11,see also 12]. The country shares five percent of the global population, but holds 35–50 percent of global civilian owned firearms [13].Three in ten US adults own a gun, about half of the non-gun owners could see themselves owning a gun in the future, and seven

in ten Americans have fired a gun [14,15]. During the Covid-19 pandemic gun purchases further increased by an estimated 64% [16], see also [17].

The percentage of gun owners is higher among white men and people in rural areas, with 72% of rural Americans growing up in a gun-household, compared to 39% in cities [15]. Stroud [18] finds that the vast majority of applicants for concealed carry licenses are white men and their primary motivation for the license is racial anxiety. Guns are more common in Republican households, with 57% of Republicans reporting a gun at home, versus 25% of Democrats, see [15]. However, between 2019 and 2021 studies find a disproportionate increase in previously underrepresented groups becoming gun owners, including women and people of color [19].

Research on gun related deaths shows that the US also has some of the highest rates of firearm deaths among high-income countries, while firearm morbidity among young people is on the rise [20–23]. In 2021, guns accounted for nearly 19% of deaths among American children and teens – surpassing car accidents, cancer, and all other causes of death among those 18 and younger [1,24], see also [25].

Yet, studies highlight that gun culture cannot be reduced to gun ownership or gun legislation; guns also have a different meaning for Americans than they do for residents of other countries [11,18,26]. Studies therefore call for more research on gun culture: the meaning people assign to guns in culture, what produces culture, and the outcomes culture has [4,6,27]. For instance, while Canadians also own many firearms, most of these are long guns, commonly used for hunting, while Americans own many more hand guns, commonly used for self-defense [26]. Scholars find many Americans see owning guns as an essential part of the country's heritage and national values. They also associate gun ownership more strongly with freedom. For many, the idea of owning guns is tied back to the second amendment of the US constitution (1791), which states that a "well-regulated militia" is necessary for the security of a free state and, therefore, the "right of the people to keep and bear Arms, shall not be infringed."

Scholars highlight that gun ownership in the US is normative – not deviant, as in other countries [28]. Most Americans own guns for leisure purposes with the main motivations being escapism (to relax and relieve stress), social interaction, and physical exercise [28], see also [29]. However, Yamane [6,30] points out that starting in the 2000s, a gun culture 2.0 has emerged in the US, in which citizens own guns as part of their right for self-defense [see also [31]. Especially politically conservative people, people with lower educational degrees, and white people who perceive a threat of economic insecurity feel empowered by guns [4]. Thereby, guns are vital elements in meaning-making and life satisfaction. Research highlights that meaning-making and belonging are primary human motivations that increase not just psychological well-being, but also resilience and life satisfaction [32], see also [33].

Scholars also point out that gun culture not only comprises meanings and stories, but also practices: clubs, training classes, collectors, and shooting associations [6]. They highlight that more research exploring gun culture in its various dimensions, meanings, and practices is needed. Understanding gun culture can also inform our understanding of culture in action more generally.

## Studying gun culture and school shootings

Given that the US has high levels of gun availability and high numbers of rampage school shootings, scholars assume that gun access plays a role in school shootings [7,34]. Yet, studies rarely examine if gun culture, especially meanings and practices around guns, mattered for shooters or if school shooters' show distinct patterns around gun culture. The concept of

culture often remains in the abstract when it comes to school shootings: Gun culture as such is a staple across the US, with millions living in the broader national gun culture. Yet, the overwhelming majority never become violent, let alone commit a school shooting: Although rampage school shootings inherently gain massive media attention, only 2% of youth firearm homicides occur at schools and only a fraction of these are rampage school shootings [25]. Mass shootings, meaning events in which four or more victims are killed across a venue (workplace, school, or other, 17), only represent 1% of gun violence in America; and among these only 25.1% occur at school [35,36].

Scholars highlight that many school shooters, especially those who kill many victims, are suicidal, many suffering from childhood trauma, marginalization, and mental disorders [7,17,37]. Further, many shooters show distinct masculinity scripts [38,39] and are inspired by previous shooters [2,40].

Exploring the role of firearms, studies on school shootings in the 1990s suggest that many shooters come from a hunting background [7], but other scholars find that firearm use does not appear to be an important factor in school shooters' surroundings [41,42]. Studies on various types of mass shootings underline the ease with which shooters obtain firearms and the strong affection many hold for their guns [17,36]. Yet, when analyzing gun legislation and mass shootings, studies do not find a clear connection: Webster et al [43] find that handgun purchaser licensing laws and bans of large-capacity magazines are associated with significant reductions in the incidence of mass shootings. However, gun legislation – such as comprehensive background checks, or assault weapons bans, which are often perceived as solutions to mass shootings – appear unrelated to fatal mass shootings.

In short, more research is needed on what gun culture in action looks like across rampage school shootings and whether school shooters show any patterns regarding gun culture. A first question is, if gun culture – meaning-making, practices, and gun access – actually matters in shooters' lives: Did shooters simply live in a country with a national gun culture in which guns being widespread is a fact of life that affects everyone as much as it did the shooters? Or do shooters show distinct meanings and practices around guns that are relevant in their social interactions? Second, does gun culture in action – the way interactional meaning making shapes practices around guns – matter for shooters' gun access?

## Materials and methods

### Outcome & sampling

To answer these questions, research needs to address the role of gun culture, not just in prominent school shootings, but ideally across all shootings. To do so, I study gun culture across a full sample of school shootings, i.e., all random active shooter attacks at school. The sample thereby includes mass shootings, as well as shootings with fewer victims and incidents during which no one was killed or injured. Mass shootings (four or more killed) cover the most gruesome and most "successful" cases from the perspective of the perpetrator, but school shootings with less victims are much more common than those resulting in mass murder [1,44]. To understand the full range of the phenomenon, research needs to include every case where a current or former student went to their school to randomly fire at people, even if they were stopped shortly after. Research points out that attacks often tend to fail due to situational dynamics [44] and that the speed of medical interventions (e.g., distance of the school to the next hospital) influences how many victims die and whether an attack results in a mass shooting. Research also shows all types of school shooting attacks lead to horrors for survivors and cause trauma across the US school system, which further increases the relevance of including ultimately failed attempts [2].

To my knowledge, no study to date comparatively analyzes a full sample of rampage school shootings. Thus, it remains unclear how representative the discussed cases and findings of previous studies are. For instance, the shootings included in prior studies may have been selected for better data availability. These are often cases with higher victim numbers (such as cases resulting in mass murder) and thus may be cases with easier access to more powerful weapons and therefore not representative of the majority of school shootings.

I define a rampage school shooting as a shooting carried out by a current or former student, at an educational facility or on its grounds, and involving a firearm and multiple victims, at least some of whom were shot randomly. This description means the perpetrator intentionally shot victims, but these victims were not previously connected to the perpetrator, for instance, they were not targeted specifically for who they are (such as an ex-partner or a teacher who gave the shooter a bad grade). I included shootings that took place in elementary, middle, and high schools, as well as colleges and universities. Some of the attacks qualify as mass shootings in which four or more victims died. My definition excludes gang related shootings, targeted revenge shootings, and shootings in which the perpetrator never attended the school.

A full set of cases that applies to this sampling frame was collected from five comprehensive databases: the *Secret Service* and *Department of Education* database [45], the *Center for Homeland Defense and Security* database [46], the FBI database [47], as well as the databases by researcher David Riedman [48] and by the ALERRT research group [49]. The full sample that covers all cases in US history comprises 83 cases. The first known rampage school shooting in the US took place in 1966. Data collection ends effective January 1, 2024.

## Data and analyses

For each case, I analyzed data from court proceedings and the police, alongside news media coverage, autopsy reports, and other scholars' assessments. I also analyzed the perpetrators' social media appearances and videotaped interrogations. I triangulated sources to verify findings. For example: did numerous independent sources suggest the shooter was a gun enthusiast and obtained the firearm legally? I assembled a case file for each shooter, comprising key information on the case including the role of gun access and culture. Data were collected over the course of several years with the help of four research assistants who did independent checks of coding decisions.

In addition to triangulating data, I also triangulated analytic approaches. I used in depth-qualitative analysis and cross-case comparisons to assess the role and possible patterns of gun culture in shooters' lives and descriptive statistics to study gun availability across all cases [50,51].

During analysis, I asked the following questions, specifying the above-outlined gaps: (1) Does gun culture play a role in shooters' lives? Meaning, did guns hold specific meaning for shooters and their social surroundings prior to the shooting, and were they part of specific practices? If so, was the gun an object of leisure, self-defense, or did it hold another meaning in the shooter's social circle? (2) Do patterns differ between younger shooters and older shooters? Younger and older shooters may show different meanings and practices around guns due to their embeddedness in familiar structures and their legal options for firearm access. (3) Do meanings and cultural practices around guns impact how shooters gained access to firearms used for the shootings? Specifically: Was gun access easy or difficult?

## Results

Findings from my qualitative in-depth analyses, cross-case comparisons, and descriptive statistics indicate that gun culture in action, as a set of meanings and practices in interactions of

shooters with their surroundings, plays a vital role in school shooters' lives prior to the shooting. I operationalize the broader concept of gun culture leaning on Swidler's [5] sociology of culture, defining culture as "symbolic vehicles of meaning" that include beliefs and informal cultural practices, stories and rituals of daily life. Importantly, culture in this view is dynamic and constantly shaped through social interactions with others.

My analysis suggests that shooters' social surroundings, families, friends, and local communities show specific meaning-making and practices around guns. Firearms were prevalent in many shooters' interactions from a young age. They were tied to practices of bonding and family life. As an important leisure item during social interactions, firearms also were a normalized household item in many of the shooters' families.

In the following sections, findings from my in-depth case analyses and cross case comparisons will carve out meanings and practices around firearms among school shooters. Although specific meanings differ in younger and older shooters, for the majority of shooters, guns played a vital role in their lives prior to the shooting (see below, for details see also S2 Data). Relying on descriptive statistics, a last section will discuss how culture in action shapes practices, showing that *all* US school shooters had easy gun access.

### Growing up with guns – US gun culture in action

Three case vignettes can illustrate the meaning-making and practices around guns. Take Adam, who was from a family of gun-enthusiasts in Arkansas. His father, who taught him how to shoot from a young age, said: "We started buying him popper guns from day one. He worked his way up to BB's and then rifles and pistols" [data source D1, see S1 Data]. When he was younger, his grandfather said, Adam had wanted to play football or basketball, but was too slight for one, too short for the other. His parents ran a local gun club and shooting was what he did best. Guns were so vital to Adam's understanding of family, that he drew guns on a school project to design a family coat of arms. His teacher later remarked she had thought it was cute because it showed the family bonded over hunting [D2]. Numerous family pictures from the 1990s show the small child with guns in his hands. In the earliest picture (see rendered sketch, Fig 1, to the left), Adam is a toddler, sitting on an eggshell-colored step in front of a white background [D 3]. He is dressed in military camouflage gear with shiny brown boots. Smiling into the camera, he has a toy gun larger than himself in his hand. Guns were so prevalent in Adam's home that many family memories revolved around them for the first decade of Adam's life – until the only 11-year tapped into his family's arsenal: He took thirteen fully loaded firearms, including three semi-automatic rifles and 200 rounds of ammunition and went on a rampage. After 10 years in prison, Adam was released on his 21st birthday. A year after his release, he applied for a concealed weapon permit.

Other shooters share similar family memories around, and fascinations with, guns as a prominent leisure activity. They too see guns as an object of great admiration, affection, and comfort, typically tied to bonding time with family. Christopher grew up around firearms that his father owned. He shot airsoft guns with his dad from an early age. A family photo [D4] from the 1990s shows Christopher on a porch, as a young boy (see rendered sketch, Fig 1, middle). He is wearing a brown shirt and jeans. In his hand, leaning on his right shoulder, an assault rifle points away from the camera. Christopher looks over his shoulder into the camera, half smiling, half looking coy. When asked in his interrogation after the shooting if he shot guns a lot and whether he was a good shot, Christopher, who had killed his father before going on his rampage, replied: "Yeah, my dad just a couple of days ago said I was a better shot than him. And he shoots a ton." The interrogator asked: "What's your favorite gun to shoot?" Christopher replied: "Couple of days ago I shot an M16, fully automatic. Probably the funnest thing I ever done" [D5, p.21]. Gushing about specific types of firearms just hours

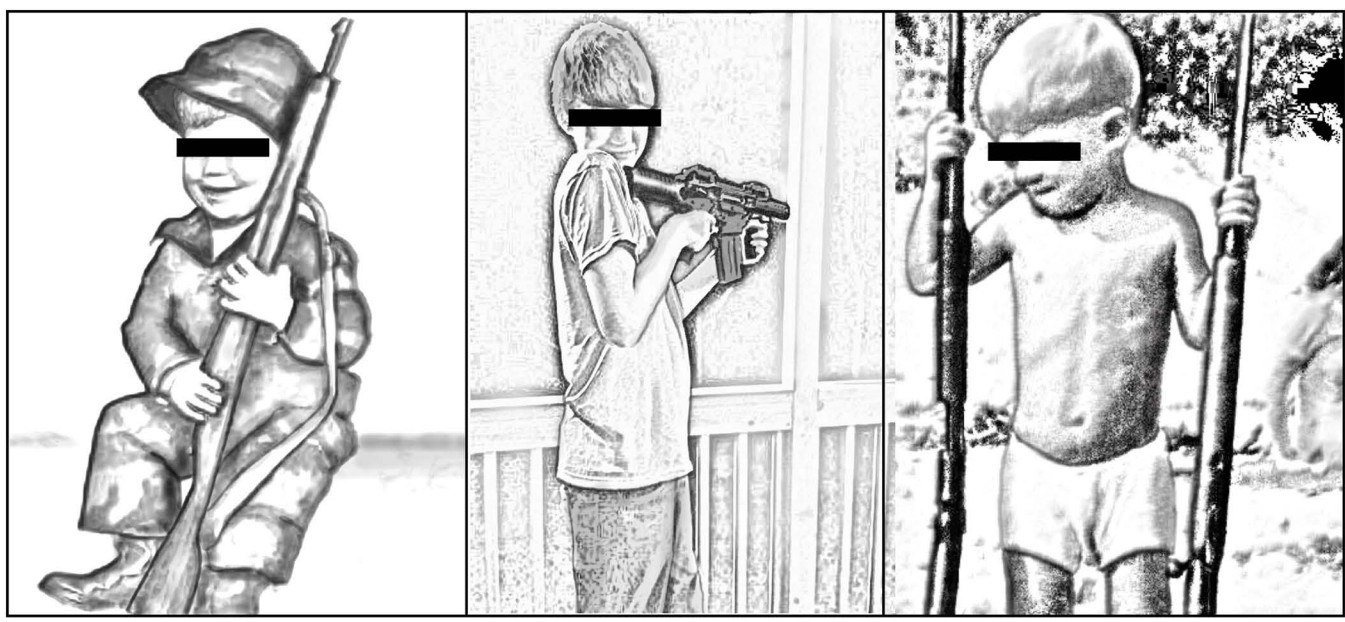

**Fig 1. Growing up with Guns.** The figure shows three school shooters as children. To maintain anonymity and prevent the perpetrator from gaining notoriety, the images are sketches based on family photographs and include a black bar over the eyes. The figure is for illustrative purposes only [original photographs are documented in sources [51,52,53].

after he killed one and injured three people, might seem odd, but for Christopher firearms were just a main family activity and a normal hobby – and being excited about their hobby is what 14-year-olds do. His statement suggests that, to him, the connection of having such easy access to firearms as a leisure activity to bond with his family and having used one of these firearms in his rampage, in which he also killed his father, was not obvious or problematic to him.

While Adam committed his attack in the 1990s and Christopher his rampage in the 2000s, Jim committed his rampage in the 1960s. Nevertheless, his growing up with guns shows similarities to Adam and Christopher, reflecting a pattern in school shooters' lives around gun culture. Jim grew up in Texas in the 1940s, his father an avid gun lover. The father trained him to shoot from a very early age and took him on hunting trips [D6]. Jim had many talents: he showed early mastery of piano, and – at the time – he was the youngest Eagle Scout in Boy Scout history. However, his father was overly critical with him in these areas of life. What impressed him and what was important to him, was how well his son could shoot [D7]. Again, family pictures tell the story: A photograph (Fig 1, right) shows Jim with his family on the beach. He is a toddler, about three years old, just able to walk around. Standing on the sand during a sunny day, Jim is wearing underpants. In each hand, he holds a rifle larger than himself, seemingly leaning on both simultaneously. He looks toward the ground, smiling at a dog out of frame [D8].

Findings across cases, illustrated here with pictures of three perpetrators, are not just illustrative of their growing up around guns. A cross-case analysis of gun culture specifics in each case, highlights that many came from a background where guns were a vital leisure activity (for details, see below; for a summary of the data for each case, see S2 Data). The three case vignettes also underline how gun culture impacts perpetrators across decades. They illustrate that shooters come from a distinct pocket of US society in which guns as leisure are part of everyday interactional practices. Here, gun culture influences actions by shaping habits,

skills, and styles by which people construct strategies of action. Culture thereby shapes social practices – what people do: In the three examples young children grew up in a distinct gun culture that led to guns being so common that perpetrators were familiar with them since they could walk. Guns were a regular item of their household, as mundane as Legos or PlayStations in other homes. It was important to their fathers that they could shoot well and they bonded with family over firearm use. As a crucial family hobby, firearms held meanings of identity and community.

Research finds the leisure gun culture to be more prominent before 2005 in the US; thereafter a culture of self-defense has come to dominate [6]. Yet, of those shooters where data indicate why the family of the shooter or the shooter themselves had a firearm (a total of 52 cases), the majority (41 cases, 78.8%) show a leisure gun culture context. Only five shooters show a self-defense background: two clear self-defense cases, two shooters where a self-defense background is likely but not certain from the data, and one case where the shooter got a gun for his protection due to untreated and severe schizophrenia – thus not a typical self-defense motivation. In six cases, data suggest the shooters neither grow up with a distinct gun culture nor had a prior affection for firearms.

This means, school shooters predominantly come from a leisure gun culture in which practices and meanings around guns construct them as objects of fun, bonding, and belonging. The prevalence of school shooters across time coming from a leisure gun culture context, despite an overall decrease of this context, suggests school shooters come from a distinct cultural pocket of US society.

These three case vignettes are no exception in the role of firearms in perpetrators lives, as the next sections, as well as the list of gun culture specifics in S2 Data, underline. What is seen in the pictures is also expressed in words: For some shooters, firearms were their "only friend"[D9], the "love of [their] life" [D10, p.45], or their "whole life" [D11]. For others they were "therapeutic" [D12], or the only topic that got an otherwise quiet and asocial shooter to passionately engage in a conversation [D13]. As the following sections show, most shooters did not just have easy access to a firearm at one specific point in time in their lives, but they were embedded in a distinct gun culture, where guns were either normalized as a household item, or were an active part of family life, since leisure activities centered around firearms as an item of bonding. Firearms acted as markers of meaning and values, symbolizing identity and group membership. Yet, specific meanings varied for younger and older shooters.

### Younger shooters and gun culture: "There isn't a whole lot a mother and a 16-year-old son can do together"

This section will take a closer look at the role of gun culture in the lives of younger school shooters – those under 18. In many US states, a person has to be at least 18 years of age to be able to legally purchase a long gun (e.g., rifle, shotgun) from a federally licensed dealer, and at least 21 to legally purchase a handgun.

A first group of younger shooters found guns for their shooting at the home of their parents, relatives, or friends, usually because they were poorly hidden. Take Christopher and Carter, introduced above: Carter found the gun that he used for his attack at home. He had grown up with guns. When an interrogator asked Carter in a videotaped interrogation, "Are you familiar with guns?," he replied: "I've been shooting them since I was about six." "Where did you get the gun from?" the interrogator asked. Carter casually replied "Found it at my dad's house" [D14].

Similarly, Christopher, who attacked his former school, explained how his father kept the gun on his nightstand in their South Carolina home [D5, p.7]: "He kept the alarm clock above it and the ammo under it. Everything was covered by a piece of paper. So how was he

supposed to know that I knew where it was?" With a teenager in the house, it is debatable if a piece of paper is an elaborated hiding place. The 14-year-old found the gun and used it to kill one person and injure three. These findings are in line with research showing that in all types of school violence, parents are the most common source for firearms [54] and 7% of children in the US live in homes where a firearm is stored unlocked and loaded [55,see also 56].

However, cases do not indicate that children could obtain their parents' firearm due to parental neglect for the child's wellbeing or safety. Police investigations and interrogations with parents suggest they often simply did not see anything wrong with a firearm in the house; in these cases guns represented either a bonding element with their kids or – less common – an object deemed necessary for self-defense. Take Kevin: His parents did not keep sugary snacks in the house because they are unhealthy. Thus, Kevin had to ride his bike five miles to the nearest gas station when he wanted sweets. However, while snacks were banned at home, firearms were left unguarded. When the voices in Kevin's head became louder, the 15-year-old slept with a loaded Glock under his pillow that his father had bought him. He later used his weapons to kill his parents and went on a rampage at school where he killed four and injured 25 people [D15].

For a second group of younger shooters, firearms were locked away at home, but they had access to the place where the gun was locked. For instance, Mathew's father kept his guns in a safe. He is one of the few cases in the sample where data suggests the gun was in family possession for self-defense reasons (see S2 Data): The family had moved to a city in Washington State and Mathew's father feared the neighborhood could be unsafe. The safe was locked, but Mathew knew the combination [D16]. Matthew used the gun to injure three and kill one person at his high school.

Similarly, the parents of 13-year-old Tony in Missouri had a locked gun safe, but the keys were in an ashtray in another part of the house. When police arrived at their home after the shooting, the ashtray was on the floor in front of the safe. Tony had used it to get the gun. As a third example in this group, the parents of 13-year-old Daniel had a gun safe in the basement of their home, where their son played first person shooter games. One day, while in the basement, he found the keys and took the guns to commit a shooting at his Indiana high school [D17].

A third group of underaged shooters bought guns with their parents or were given guns as a present from their parents or other close relatives for leisure purposes. For instance, Kevin (see also above) had severe mental health issues and had just started undergoing treatment with Prozac, when his father bought a 9mm Glock for him [D18]. Three months later, Kevin stopped taking Prozac and his father bought him another firearm, a semiautomatic rifle. 15-year-old Kevin used both weapons, and a third he had bought from a friend, to commit his shooting. When his psychiatrist was later asked if he had any concerns about the gun purchases, given Kevin's mental disorder and his excessive interest in guns, he said, "No one consulted me about that decision, and yes, I have concerns about that" [D18].

In the fourth and last group of younger shooters, firearms played an even more active role in parent-child bonding. Take, Jack, whose mother expressed that she saw a firearm as a way to bond with her son, who suffered from mental health issues. She knew her son would go on a hunting trip to spend time with his father and asked his psychiatrists if he thought this was okay. The psychiatrist gave the green light [D19]. However, Jack's mother feared that he would not be prepared for the trip and would be embarrassed because he could not shoot well. Therefore, she looked to buy him a shotgun for practice. In her mind, this also gave her a chance to connect with her kid, because "I had been thinking that when [Jack] was going to be doing the hunting, I figured well, a couple of times I had done trap shooting, back in the 70s, skeet shooting, and I figured, well, this could be something then, we could go target

shooting, there isn't a whole lot a mother and a 16-year-old son can do together" [D19]. She saw recreational shooting as one of the few ways to engage in mother-son-activity. She went to the store to have a look at the firearms first, without the intent to buy. However, she recalls that the sales person was very good at his job and she decided to buy the gun for her son right away. Jack later used it to commit a rampage shooting at his high school.

These exemplary cases highlight how culture in action shapes practices in systematic ways: Shooters and their social surroundings attributed affectionate meaning to firearms and constructed everyday practices of bonding around them. Many parents of underaged school shooters did not perceive it as unusual, or dangerous, to train their kid to fire weapons with live ammunition and to give them access to guns from a young age. A next-door neighbor, who considered 16-year-old Gordon to be a younger brother prior to his 2019 school shooting, said about the passionate hunting trips Gordon took with his father: "The hunting ... it's about… you ... you turn all of this off. It's just quiet. And you just… you have an opportunity to connect with your son in a different way." [D20, min 0.58]. A psychologist who interviewed 16-year-old A.J. after his attack later testified about the meanings and practices around firearms in A.J.'s life that "guns were the love of his life" [D21, p.45]. This affectionate cultural meaning of guns as family bonding and socializing qualities, also meant that, in some shooters' homes, firearms were lying around like a hairbrush – easily accessible to everybody in that home, thus providing easy means for children considering a shooting (see below).

### Older shooters, leisure, and legal firearm access: "I have done it!!! Today, I […] bought a 12-gauge pump-action shotgun!!!"

Despite their opportunities for purchasing firearms, older shooters are the smaller group in the sample: the median shooter is 17 years old and 59% of the shooters are under 18. Older shooters differ from younger shooters, not just because they can legally obtain firearms and because they are more likely to live independently (and are thus less likely to gain access through parents), but also in the symbolic meanings they assign to firearms. While firearms are highly relevant for younger shooters' belonging and group identity, for older shooters they seem more important for self-identity and well-being. To many older shooters, guns were their only interest, the only way in which they were social and had a hobby. For many, this hobby was a refuge from their anxieties, isolation, and severe mental disorders. Thus, guns positively impacted the older shooters' well-being and many formed their personal identity around them.

First, regarding access, most older shooters legally obtained guns by buying them. For instance, William ordered his semiautomatic rifle for $129 at a gun shop over the telephone, plus 200 rounds of ammunition, all delivered to his dorm where he later started his shooting [D22]. Mental health issues did not stop shooters from legally purchasing a firearm: For instance, Caleb, Joon, and Simon all bought their guns legally, despite severe mental disorders [D23, D24]. Despite previously being treated at a mental health clinic for severe mental disorders and being well known for his frustrated violent outbursts, 19-year-old Caleb legally bought an AR-15 rifle and large amounts of ammunition after passing the background check in Florida [D12]. In total he purchased five guns within a year – even though his record included a brief investigation by the FBI for making threats to commit a school shooting. Caleb's mother, although not a fan of firearms, wanted to support her son in his only passion and the family he lived with after her sudden death insisted that he had to keep the firearm in a locked cabinet, but he had the keys to the lock. It was his legal right to own a firearm and the people around him accepted that guns were the passion of his life. Firearms were the one thing that brought the isolated and mentally distraught Caleb joy and the only thing he was good at. Firearm practice and competition was where he felt he belonged and he wore his JROTC

T-Shirt with pride – even on the day of the shooting. Schoolmates recall how much he loved guns: "[He] would tell us, 'Oh, it was so fun to shoot this rifle' or 'It was so fun to shoot that.' It seemed almost therapeutic to him, the way he spoke about it" [D12]. The CCTV recording of Caleb's brother visiting Caleb in custody a few hours after his attack shows him telling Caleb that he finds it surprising that Caleb never had any interest besides firearms. Later, trying to convey to Caleb the gravity of what he had just done by killing 19 people and injuring another 19 people at his former high school, his brother tells him, "You can't even own a gun now" [D25], as the loss of these beloved items may convey to his brother the seriousness of the situation. Expressing a similar sentiment, Emmett, who committed a school shooting in 1988, told the detective after his arrest, "The only friend I had was my gun and you already took that from me" [D9].

In another example of this pattern, the mother of 20-year-old Allan tried to support her son with his mental struggles, by encouraging his main hobby: firearms. She owed the various guns he later used in the rampage, including the one Allan used to kill her. She had obtained them to go to shooting ranges with her son as a recreational activity. Having grown up with firearms and having a pistol permit, she attended NRA courses with him [D26]. With severe social anxieties, shooting firearms was the only hobby that got Allan out the house. They helped with his well-being and his mother wanted to support him in this endeavor. She had another gun scheduled as a gift for Allan for Christmas, but before Christmas came, Allan shot her and went to kill 25 people at his former elementary school.

A third case further illustrates the affectionate cultural meaning gun purchases held for older shooters. Court documents show that 18-year-old Mateo legally bought the firearm that he would later use in his North Carolina shooting for 217 USD. In his diary, he wrote, "I have done it!!! Today, I went to Wal-Mart at 2:20 p.m. and I bought a 12-gauge pump-action shotgun!!!" After further firearm purchases, he wrote: "My weapons are my lovers. I spend a lot of time with them and I hope they will not leave me. I am always faithful to them" [D27].

Research on hobbies shows that hobbies are a crucial driver of individual and collective identity formation. They positively affect mental health and well-being, allowing for a structured way to spend time [57]. Firearms therefore have an appeal as hobbies. Collins highlights that many gun owners spend much time with their firearms, even when not shooting them, regularly cleaning and reloading them [58]. But their appeal is amplified for many school shooters: With many of the older shooters feeling desperate and powerless, being marginalized and suffering from mental disorders, findings suggest firearms have a special draw, helping with anxieties and often being the one source of joy, belonging and connection to the outside world [see also 4]. Literature on gun ownership suggests overall gun owners report higher levels of alienation from society and people who feel they are misfits and people who feel economically left behind, feel empowered through guns [4,17]. Guns thereby tend to create a sense of belonging that most school shooters otherwise lack. Thwarted belongingness, shown to be a vital component in suicides [59,60], also seems to be a relevant component in many school shooters' lives [see also 7,39,61].

## Easy gun access and US rampage school shootings

Culture in action means that meaning making shapes behavior [5]. Findings suggest this to be the case in school shooters' lives and firearm availability: despite differences in age, mental health background, as well as the year and location of the shooting, all school shooters have easy gun access.

Relying on descriptive statistics, Table 1 illustrates gun access across the entire sample: It lists the shooter age, whether their gun access was easy or not, alongside how perpetrators obtained their firearms. I distinguish four categories of gun access: "very easy," "easy,"

**Table 1. Descriptive Statistics.**

| Variable | | | | | | |
|---|---|---|---|---|---|---|
| *Age of shooter* | *min* | *1st quartile* | *median* | *mean* | *3rd quartile* | *max* |
| | 11 | 14.5 | 17 | 18.5 | 19 | 62 |
| *Gun Access* | | | | | | |
| | *very easy* | *easy* | *difficult* | *very difficult* | | *NA* |
| n | 66 | 15 | 0 | 0 | | 2 |
| percentile | 79.5 | 18.1 | 0 | 0 | | 2.4 |
| *Gun Origin* | *home* | *bought* | *other* | | | *NA* |
| n | 47 | 28 | 6 | | | 2 |
| percentile | 56.6 | 33.7 | 7.2 | | | 2.4 |
| n=83 | | | | | | |

"difficult," and "very difficult." "Very easy" means either shooters legally owned a gun or obtained the gun legally with ease, from the house, car, or other property of a relative, neighbor, or friend, where it was accessible committing no illegal action. "Easy" means gun access was still easy, but not very easy: although obtaining the firearm within a familiar context, perpetrators had to go through a few steps and did not have immediate access. "Difficult" means that perpetrators had to take several steps to obtain a firearm, organize a complex purchase over a prolonged period of time, and / or operate outside a familiar context (such as friends or family). "Very difficult" means they had to go through great lengths and do something that would result in severe social, or legal punishment. The distinction follows the fuzzy set QCA logic of a factor being fully in, fully out, more in than out, and more out than in of a set [see also reference [62] for qualitative coding using fuzzy set borders]: very easy (1.0), easy (0.7), difficult (0.3) and very difficult (0).[Table 1 about here]

Table 1 shows that most shooters were too young to legally buy a firearm, with a median age of 17 years. Nevertheless, across *all* US school shootings, gun access was easy: It was either very easy (79.5%), or easy (18.1%). Those with "very easy gun access" took the gun from a familiar context or obtained it legally. They were given the gun as a present, they bought the gun at a store, or they simply took the gun from their parents' bedroom, kitchen cabinet, or a relatives' closet. The remaining 18.1% of all school shooters showed "easy gun access," meaning gun access was still easy, but not "very easy" (for one case data does not indicate whether the shooter's gun access was easy or not easy; it is unclear how he obtained the firearm). For example, Ewan and Dawson were not able to legally purchase a gun and went to a gun show with a friend who was 18 years old and thereby legally old enough to do so [D28]. Again, while access here does not qualify as "very easy," as it required asking a friend to purchase the gun, gun access was still "easy:" it was fast and occurred in a familiar context.

Gun access splits the group between the shooters 18 and older and younger than 18. Those 18 and older show mostly legal purchases. Only a few of the older shooters obtained their guns from their family home. Those younger than 18 mostly show gun access from home. But, as Table 1 shows, although the median age of shooters was 17 – thus most were not legally allowed to buy firearms – they still had easy access. A majority (56.6%) of shooters took the firearm from home, 33.7% bought their guns legally, 7.2% obtained them through other sources, and two shooters (2,4%) obtained them from unknown sources (for details, see S2 Data). Previous sections underlined the role firearms played in family bonding for younger shooters, suggesting this meaning making translates into cultural practices of firearm availability.

Interestingly, there is not a single rampage school shooting in the US in which shooters faced "difficult" or "very difficult" gun access, despite some of the shooters being only 11 or 13 years old. Research points out that many school shooters in the US did not show commonly identified factors for school shootings, such as marginalization or mental disorders][7]. Yet, findings suggest this is not the case for easy access to firearms: "easy" and "very easy" gun access covers *all* school shootings.

Thus, findings suggest that if access to firearms is even somewhat blocked, shooters may be deterred. This is also visible in the guns used in the shootings. For instance, police statements show Carter wanted to get more powerful weapons for his rampage, but these were locked away [D29]. He then injured one person at school with the only gun he could easily obtain. Christopher also later lamented that he only had easy access to – and therefore only took – the pistol his father kept in the dresser drawer, while the assault rifle was in the safe [D5].

The temporal analysis shows that between 1960 and 1992, obtaining the firearm from home was the exception (only one of 11 shooters during this timeframe, a shooter in 1985, obtained his gun from home). Yet, between 1993 and 2005, the situation switched: Legally obtained guns were the exception (only two of 24 shooters). This coincides with major changes in US gun legislation: In the mid-1990s, firearms access was restricted via legislation. One of the main pieces of legislation, the Public Safety and Recreational Firearms Use Protection Act from September 1994 expired in September 2004. As Fig 2 illustrates, during the ban (grey), only two weapons were legally obtained by shooters; immediately after it expired, shooters regularly used purchased firearms. Still, findings suggest shootings did not decline during the ban, they only show a dip in the early 2000s. Thus, even as the legal frameworks change, access is not thwarted, it merely switches to the private sphere, where guns are very prevalent among the group of school shooters.

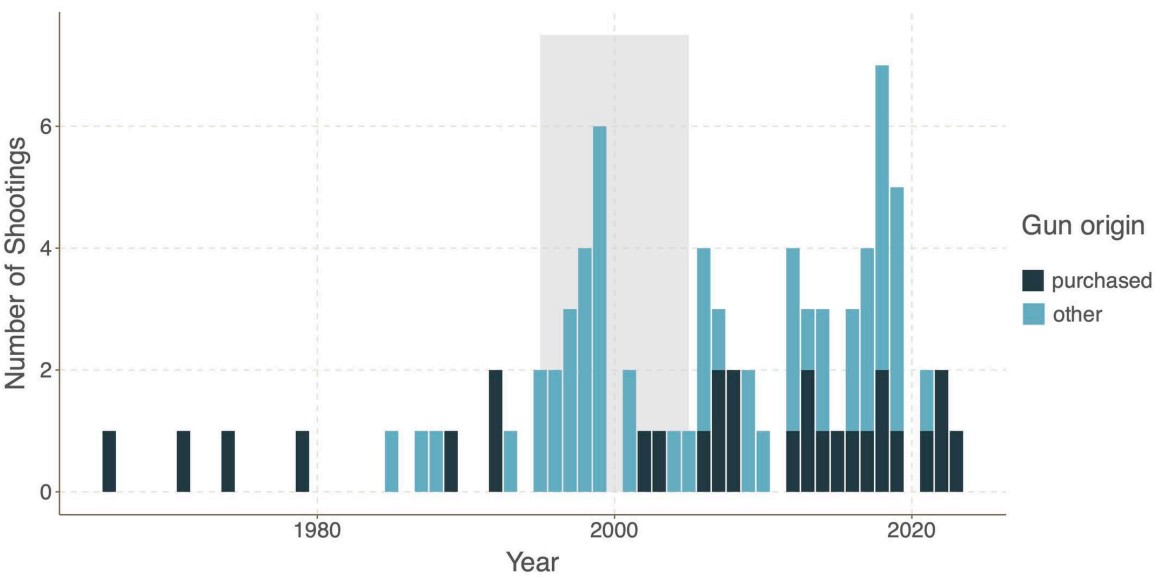

Note. The grey area marks the duration of the Public Safety and Recreational Firearms Use Protection Act (09/1994 - 09/2004).

**Fig 2. Gun Access Over Time.**

## Discussion and conclusions

This article explores the role of gun culture – meaning making, practices, and access to firearms – in rampage school shootings using in-depth case analyses, cross case comparisons, and descriptive statistics of a full sample of school shootings. Findings suggest that many school shooters grew up with guns as an important part of their social lives, filled with distinct meanings of bonding and attached to vital social practices. Younger shooters show a strong relevance of firearms for belonging and group identity, while for older shooters, guns show a stronger relevance for self-identity and well-being. For many shooters, guns are their only hobby, whether the solitary source of joy and calm, or the key way shooters interact with others – through talks about firearms or going shooting with others. Findings suggest culture translates into practices and shapes human behavior: The distinct gun culture and meaning-making around firearms among school shooters coincides with easy access across cases.

Gun culture in action is thus visible across most school shooters' lives. Findings highlight that the abstract concept of gun culture can be operationalized and tracked in empirical data and that the unique US gun culture seems connected to the unique phenomenon of US school shootings. The finding that no shooter had difficulties obtaining a firearm, suggests that easy gun access is a vital component in, and may be a necessary condition for, school shootings (all shootings show this condition, while not all shootings show other context factors, like marginalization or mental disorders).

The finding that school shooters come from a distinct cultural pocket in the US, a leisure gun culture that is otherwise declining, narrows down the group of possible perpetrators. Yet, only a very small sub-group of Americans with easy gun access go on a rampage. Therefore, a limitation of this study – and others in the field – is the lack of negative comparison cases: More research is needed on students with the same context factors such as marginalization or family trauma, as well as easy gun access, who never commit a school shooting. Moreover, future research should examine whether this gun culture in action only describes a shooter profile, or whether it also holds relevance for the shooters' decision to commit a shooting. More insights are needed whether those who contemplate committing a shooting but do *not* have easy access to firearms decide against a shooting, or if easy access gives shooters the idea to commit a shooting.

Research also needs to reassess prevention measures, since none of the current measures clearly diminish firearm violence in schools [25]. First, findings underline the role of meaning making and belonging in school shooters' lives. With belonging being vital for human well-being and life satisfaction [32], findings call for more research on prevention measures fostering student belonging and thereby possibly reducing violent behavior overall [17], see also [63, 64].

Second, findings highlight the relevance of safe storage for prevention. Findings call into question the notion that motivated perpetrators could obtain a firearm either way; de facto, not a single shooter even faced the decision to illegally obtain a firearm. In every single rampage school shooting in the US, shooters had easy access to the weapon they used. With harsher gun legislation, there was a strong decline in school shooters obtaining their gun legally, suggesting gun legislation affects gun access. Yet, shootings did not decrease during this timeframe, nor did they increase after the ban ended; shooters simply took guns from home while the ban was in place and legally acquired them thereafter. Findings suggest this is because guns are already widespread in the US and normalized in pockets of US society.

Further, findings suggest that to prevent school shootings, not only must the prevalence of guns and legal access for potential shooters over 18 be considered, but also easy unguarded access for children at home. More research is needed on how guns can be stored safely away

from children and young adults. With research indicating that gun ownership is generally increasing, including among previously underrepresented groups, the relevance of safe storage will further increase. Between 2002 and 2015, the number of US children living in homes with at least one unlocked and loaded firearm has increased from about 1.6 million to 4.6 million [55]. Research highlights that safe storage practices – e.g., storing firearms and ammunition separately – can decrease the risk of firearm deaths among children [65,36,see also 66]. Findings suggest not having a firearm at home, or if a firearm is stored, storing it safely, may not just help prevent accidents and suicides with firearms at home, but also planned attacks, like school shootings. Further, the often implemented security measures at school do not affect the great majority of firearm deaths (homicides, suicides and accidents) that occur outside of school [25], while gun access might aid prevention in both areas.

Third, findings underline the role of parents or other adults in gun access. Among all rampage school shooters under 18 – the minimum legal age to purchase a firearm in most US States – none showed difficulties obtaining a firearm. This may indicate that those who face difficulty in obtaining a firearm simply do not commit a school shooting. Thereby, findings emphasize that avoiding access through adults, especially parents, is vital for prevention. 2024 marked the first time in history that a lawsuit held the parents of a school shooter accountable. The parents of 15-year-old Billy, who attacked his Michigan school, were convicted of manslaughter and sentenced to 10–15 years in prison due to their son's "unfettered access to a gun or guns as well as ammunition in [their] home" and since they "glorified the use and possession of these weapons," according to the judge [67]. These two aspects mentioned by the judge directly correspond to meaning making around, alongside access to, firearms, as discussed in this article. In light of these findings, helping parents understand what safe storage means and holding them accountable for unsafe storage are worth exploring as avenues for effective prevention. However, a positive effect of longer prison sentences for parents, as in Billy's case, can be called into question given that research suggests that longer prison sentences do not affect crime deterrence or reoffending [see 68,69].

Even shooters themselves often feel they should not have been able to legally obtain a firearm. "Do you feel that you should have had a gun?" a reporter asked Liam right after his shooting. Liam replies casually and sincerely: "No, sir." Before attacking his university in 2008, Simon, who suffered from psychosis, wrote an essay laying out why people with mental disorders, like himself, should not be able to obtain firearms so easily. Jack, who committed his shooting in 2004, aged 16, started advocating for gun safety from prison and supports stricter gun legislation. Noah, who injured two at his Virginia community college in 2013, now expresses a similar stance on gun access. While discussing his plans to commit the shooting, the 18-year-old discussed in detail how he thought it was too easy for individuals in the US to obtain firearms [70]: "I'm not saying we need cops in schools, but we need to have something where people can't hurt people. It's just real easy to get guns."

## Supporting information

**S1 Data. Cited data sources.** This supporting information lists all sources referenced in the article. The list does not include all analyzed sources for this project, as these by far exceed those listed here. It includes sources that are directly referenced in the article when discussing cases as examples of larger identified patterns.
(PDF)

**S2 Data. Full dataset.** This table summarizes key data points from the analysis in numerical form and provides information from the in-depth case analyses. Whether gun access was easy or not was coded as fuzzy sets: very easy = 1; easy = 0.7; difficult = 0.3; very difficult =0. Access

of the gun was coded as 1 = from home, 2 = legally bought, 3 = other. The notes provide a brief summary on gun culture, access, and relevance of the gun in the shooters' life for each case.
(PDF)

## Acknowledgements

I would like to thank Nicolas M. Legewie for his help with the analysis relying on descriptive statistics, with Fig 2, and for his feedback on my larger project on school shootings. I also thank my research assistants, Katarina Dacić, Charlie Zaharoff, Valentin Ahles, and Richard Lehman, for their help in collecting data, fact-checking information, and double-checking codes. I would like to thank the Open Access Fund at University of Erfurt for their support in covering part of the open access publication costs of this article.

## Author contributions

**Conceptualization:** Anne Nassauer.

**Data curation:** Anne Nassauer.

**Formal analysis:** Anne Nassauer.

**Investigation:** Anne Nassauer.

**Methodology:** Anne Nassauer.

**Project administration:** Anne Nassauer.

**Supervision:** Anne Nassauer.

**Validation:** Anne Nassauer.

**Writing – original draft:** Anne Nassauer.

**Writing – review & editing:** Anne Nassauer.

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
