## [Decision Letter · Decision Letter 0]

11 Dec 2024

PONE-D-24-36435“The only friend I had was my gun”: A mixed-methods study of gun culture in school shootingsPLOS ONE

Dear Dr. Nassauer,

Thank you for submitting your manuscript to PLOS ONE. After careful consideration, we feel that it has merit but does not fully meet PLOS ONE’s publication criteria as it currently stands. Therefore, we invite you to submit a revised version of the manuscript that addresses the points raised during the review process.

We look forward to receiving your revised manuscript.

Kind regards,

James C. Wo, Ph.D.

Academic Editor

PLOS ONE

Journal Requirements:

2. Please include a separate caption for each figure in your manuscript.

3. We note that Figure 2 in your submission contain copyrighted images. All PLOS content is published under the Creative Commons Attribution License (CC BY 4.0), which means that the manuscript, images, and Supporting Information files will be freely available online, and any third party is permitted to access, download, copy, distribute, and use these materials in any way, even commercially, with proper attribution. For more information, see our copyright guidelines: http://journals.plos.org/plosone/s/licenses-and-copyright.

1) You may seek permission from the original copyright holder of Figure 2 to publish the content specifically under the CC BY 4.0 license. 

2) If you are unable to obtain permission from the original copyright holder to publish these figures under the CC BY 4.0 license or if the copyright holder’s requirements are incompatible with the CC BY 4.0 license, please either i) remove the figure or ii) supply a replacement figure that complies with the CC BY 4.0 license. Please check copyright information on all replacement figures and update the figure caption with source information. 

If applicable, please specify in the figure caption text when a figure is similar but not identical to the original image and is therefore for illustrative purposes only.

4. We note that there is identifying data in the Supporting Information file <06_Appendix B. Data.docx>. Due to the inclusion of these potentially identifying data, we have removed this file from your file inventory. Prior to sharing human research participant data, authors should consult with an ethics committee to ensure data are shared in accordance with participant consent and all applicable local laws.

-Location data

Please remove or anonymize all personal information (Shooting date, Shooting age), ensure that the data shared are in accordance with participant consent, and re-upload a fully anonymized data set. Please note that spreadsheet columns with personal information must be removed and not hidden as all hidden columns will appear in the published file.

**Additional Editor Comments:**

The two reviewers conclude that there is merit to this paper; however, both of them (including myself) have deemed the paper to be undertheorized. Specifically, both reviewers call on the author to more effectively link their analysis and findings to the robust extant literature on guns and gun culture. Relevant theoretical and empirical work was either ignored or the author was unaware of them. Additionally, the paper needs to more effectively highlight the contributions of the current study, as some of the findings seemingly have already been found by others. I want to be direct with the author; for your paper to proceed to publication; you will need to make significant revisions to address the concerns from the two reviewers, as well as my overarching concerns with ineffective theory and the impact of the findings to the extant literature.

Reviewers' comments:

Reviewer's Responses to Questions

**Comments to the Author**

1. Is the manuscript technically sound, and do the data support the conclusions?

Reviewer #1: Partly

Reviewer #2: Partly

2. Has the statistical analysis been performed appropriately and rigorously? 

Reviewer #1: N/A

Reviewer #2: N/A

3. Have the authors made all data underlying the findings in their manuscript fully available?

Reviewer #1: Yes

Reviewer #2: Yes

4. Is the manuscript presented in an intelligible fashion and written in standard English?

Reviewer #1: Yes

Reviewer #2: Yes

5. Review Comments to the Author

Reviewer #1: I have submitted my review as an attached document. For my review, please see attached document for my comments. In short, I believe this paper needs major revisions, foremost because multiple crucial studies that have been done in this space in recent years or that are relevant to this space are uncited in this paper, which make me suspect that the author is unfamiliar with this relevant prior work.

Reviewer #2: The researcher conducts qualitative analyses and cross-case comparisons to examine patterns in gun culture related to school shootings and explore culture “as a process shaped by individuals and their social environments through everyday practices and interactions, rather than as a set of static beliefs”. The author uses a sample of all US rampage school shootings = “shooting carried out by a current or former student, at an educational facility or on its grounds, and involving a firearm and multiple victims, at least some

of whom were shot randomly”. The full sample that covers all cases in US history consists of 83 cases, from 1966 to 2023.

The author aimed to answer the following questions: Does gun culture play a role in shooters’ lives? The answer is yes! Firearms were prevalent in many shooters’ interactions from a young age. They were tied to practices of bonding and family life

(2) Did cultural practices around guns impact how shooters gained access to firearms used for the shootings? (3) Do patterns differ in younger shooters and older shooters? Younger and older shooters may show different practices and meanings around guns due to their embeddedness in familiar structures and their legal options for access to a firearm.

The article is interesting and reports several aspects of the meanings and practices around guns in shooters' lives, using descriptive statistics to track how shooters acquire guns, for example. The paper is descriptive and has no link to any theory; for instance, why was the routine activity approach not adopted to explain some of the situational aspects of the shootings?

The article can be improved and here are my suggestions:

1) Include all details of the analysis, some quantitative, into the methods and delete them from the footnote. Example footnote 3.

2) Keep footnotes to a minimum.

3) If the author has had access to information on situational conditions about the shooting, why was this discussed? Distance from the residence of the shooter to the school, type of neighbourhood, close bars, desolate place? See

Ceccato, V., & Westman, J. (2024). Where Does Firearm-Related Violence Occur in Cities?. Journal of Planning Literature, 08854122231219918.The paper discusses relevant land use and POI for shootings. The presence of schools was actually protective (except when the school was the target)

4) The temporal dimension is a crucial aspect. The kids had to be in school. No reference to the spatial-temporal aspects of the shootings.

5) Too long title - This subheading is too long and difficult to understand.

Younger Shooters and Gun Culture Access: “There isn’t a whole lot a mother and a 16-year old son can do together”

6) Discussion –

How does the author land on this conclusion? It is not clear

“With harsher gun legislation, we see a strong decline in school shooters obtaining their gun legally, suggesting gun legislation matters for gun access. “

7) The discussion section is too short. Create a section for “conclusions and recommendations”.

6. PLOS authors have the option to publish the peer review history of their article (what does this mean? ). If published, this will include your full peer review and any attached files.

**Do you want your identity to be public for this peer review?** For information about this choice, including consent withdrawal, please see our Privacy Policy .

Reviewer #1: No

Reviewer #2: No

---

## [Author Response · Author response to Decision Letter 1]

10 Feb 2025

I want to express my sincere appreciation to the editor and the reviewer for their careful reading and these thoughtful and constructive comments. I addressed all comments in my revision. I have also changed the entire manuscript formatting according to the PLOS ONE Guidelines, changed the figure captions and have replaced the pictures in figure 1 with drawings. Please see my document "Response to Reviewers" for details.

---

## [Decision Letter · Decision Letter 1]

19 Mar 2025

“The only friend I had was my gun”: A mixed-methods study of gun culture in school shootings

PONE-D-24-36435R1

Dear Dr. Nassauer,

We’re pleased to inform you that your manuscript has been judged scientifically suitable for publication and will be formally accepted for publication once it meets all outstanding technical requirements.

Kind regards,

James C. Wo, Ph.D.

Academic Editor

PLOS ONE

Additional Editor Comments (optional):

The reviewer appreciated the significant changes you made and so did I.

Reviewers' comments:

Reviewer's Responses to Questions

**Comments to the Author**

1. If the authors have adequately addressed your comments raised in a previous round of review and you feel that this manuscript is now acceptable for publication, you may indicate that here to bypass the “Comments to the Author” section, enter your conflict of interest statement in the “Confidential to Editor” section, and submit your "Accept" recommendation.

Reviewer #1: (No Response)

2. Is the manuscript technically sound, and do the data support the conclusions?

Reviewer #1: Yes

3. Has the statistical analysis been performed appropriately and rigorously? 

Reviewer #1: Yes

4. Have the authors made all data underlying the findings in their manuscript fully available?

Reviewer #1: Yes

5. Is the manuscript presented in an intelligible fashion and written in standard English?

Reviewer #1: Yes

6. Review Comments to the Author

Reviewer #1: This turned into a great paper in this revised version. I enjoyed reading it.

There are some minor typos (e.g. references in the text saying "Author 2024"; and in one place you wrote "familiarities" when you probably meant "similarities") that should be corrected when proofing the final version.

7. PLOS authors have the option to publish the peer review history of their article (what does this mean? ). If published, this will include your full peer review and any attached files.

**Do you want your identity to be public for this peer review?** For information about this choice, including consent withdrawal, please see our Privacy Policy .

Reviewer #1: **Yes: ** Patricia Jewett

---

## [Editor Report · Acceptance letter]

PONE-D-24-36435R1

PLOS ONE

Dear Dr. Nassauer,

I'm pleased to inform you that your manuscript has been deemed suitable for publication in PLOS ONE. Congratulations! Your manuscript is now being handed over to our production team.

Kind regards,

on behalf of

Dr. James C. Wo

Academic Editor

PLOS ONE